Modified generalized method of moments for a robust estimation of polytomous logistic model

Wang Xiaoshan xwang@forsyth.org
Department of Clinical and Translational Research/Forsyth Institute , Cambridge, MA , USA
Department of Oral Health Policy and Epidemiology, Harvard School of Dental Medicine , Cambridge, MA , USA
Emmert-Streib Frank
Electronic publication date: 2014 Jul 1
Publication date: 2014
Volume: 2
Electronic Location ID: e467
Received 2013 Dec 11; Accepted 2014 Jun 13
Copyright: © 2014 Wang
Copyright year: 2014
Copyright holder: Wang
License: This is an open access article distributed under the terms of the Creative Commons Attribution License, which permits unrestricted use, distribution, and reproduction in any medium, provided the original author and source are credited.
License URL: https://creativecommons.org/licenses/by/3.0/

Keywords: Robust statistics, Generalized method of weighted moments, Polytomous logistic model

Funding: No funding was provided for this work.

==============================
The maximum likelihood estimation (MLE) method, typically used for polytomous logistic regression, is prone to bias due to both misclassification in outcome and contamination in the design matrix. Hence, robust estimators are needed. In this study, we propose such a method for nominal response data with continuous covariates. A generalized method of weighted moments (GMWM) approach is developed for dealing with contaminated polytomous response data. In this approach, distances are calculated based on individual sample moments. And Huber weights are applied to those observations with large distances. Mellow-type weights are also used to downplay leverage points. We describe theoretical properties of the proposed approach. Simulations suggest that the GMWM performs very well in correcting contamination-caused biases. An empirical application of the GMWM estimator on data from a survey demonstrates its usefulness.

Introduction

Polytomous logistic regression models for multinomial data are a powerful technique for relating dependent categorical responses to both categorical and continuous explanatory covariates (McCullagh & Nelder, 1989; Liu & Agresti, 2005). In practice, however, the model building process can be highly influenced by peculiarities in the data. The maximum likelihood estimation (MLE) method, typically used for the polytomous logistic regression model (PLRM), is prone to bias due to both misclassification in outcome and contamination in the design matrix (Pregibon, 1982; Copas, 1988). Hence, robust estimators are needed.

For categorical covariates, we may apply MGP estimator (Victoria-Feser & Ronchetti, 1997), ϕ-divergence estimator (Gupta et al., 2006), and robust quadratic distance estimator (Flores, 2001). The least quartile dfference estimator can deal with overdispersion problem (Mebane & Sekhon, 2004). But all these methods are difficult to adapt for continuous covariates.

A generalized method of moments (GMM) estimation can be formed as a substitute of MLE. The GMM is particularly useful when the moment conditions are relatively easy to obtain. GMM has been extensively studied in econometrics (Hansen, 1982; Newey & West, 1987; Pakes & Pollard, 1989; Hansen, Heaton & Yaron, 1996; Newey & McFadden, 1994). Under some regularity conditions, the GMM estimator is consistent (Hansen, 1982). With an appropriately chosen weight matrix, GMM achieves the same efficiency as the MLE (Hayashi, 2000). Furthermore, under certain circumstances, GMM provides more flexibility, such as dealing with endogeneity through instrumental variables (Baum, Schaffer & Stillman, 2002).

Like MLE, GMM estimation can be easily corrupted by aberrant observations (Ronchetti & Trojani, 2001). Such observations can bring up disastrous bias on standard parameter estimates if they are not properly accounted for, see Huber (1981), Hampel et al. (2005), and Rousseeuw & Leroy (2003). So we propose a modified estimation method based on an outlier robust variant of GMM. The method is different from the kernel-weighted GMM developed for linear time-series data by Kuersteiner (2012) in that this is a data-driven method for defining weights. The new approach is evaluated using asymptotic theory, simulations, and an empirical example.

The robust GMM estimator is motivated by the data from a 2006 study on hypertension in a sample of the Chinese population. 520 people completed the survey. Observed variables included demographics, social-economic status, weight, height, blood pressure, and food consumption. Sodium intakes were calculated based on overall food consumption. Among those covariates, age, body mass index (BMI), and sodium intakes are all continuous. Based on blood pressure measurements, subjects were classified into 4 categories: Normal, Pre-hypertension, Stage 1 and Stage 2 hypertension. Table 1 lists the summary statistics of the sample. One of the research objectives is to examine the association between hypertension and risk factors in the population. Since the proportional odds assumption is violated (Score test for the proportional odds assumption gives χ2=182.27 with a degree of freedom of 8, p<0.0001), we apply the polytomous logistic model, using the normal category as the reference level. In the case of J category, the polytomous logit model have J−1 comparisons. Each comparison have a set of parameters for all covariates in the model. Therefore, the generalized logit model is not parsimonious when comparing with the proportional odds model. But the simultaneous estimation of all parameters is more efficient than separate models for each comparison. It is another option for ordinal response data, especially when a proportional odds model does not fit the data well. Table 2 lists the output from the model estimated by MLE. It is obvious that, if MLE is used, the estimates is inconsistent for sodium intakes, particularly the negative coefficient of sodium intake for the odds between the Stage 2 hypertension and the Normal categories. The inconsistency is more obvious when we plot the odds with respect to the sodium intake, the downward trend of the odds in Fig. 2A. This result contradicts the previous finding that there is a strong relationship between sodium intake and hypertension, see for example National Research Council (2005), He & MacGregor (2004) and references therein. Besides, Fig. 2A also shows another strange situation: the higher starting points for the odds between the Pre-hypertension and the Normal categories. The scatter plot (Fig. 1) between distances and leverages suggests some observations are possible outliers: Observations 21, 33, 85, 92, 194, 274, 336, 414, 459, 483, and 489 have large distances, which are blue-colored, and Observations 37, 83, 263, 459, 483, 485, and 490 have large leverages, which are red-colored.

Figure 1 Scatter plot of distance vs. leverage, which are based on MLE.

Criteria cd for the distance and cx for the leverage are demonstrated.

Figure 2 Compare odds plots of sodium intakes between MLE estimates and GMWM estimates on the population of female, age = 40, and BMI = 23.

Table 1 Summary statistics for surveyed subjects.

Covariate		Hypertension categories	
		Normal	Pre-hypertension	Stage 1	Stage 2	
Gender	Male	138	104	29	8	
	Female	87	114	31	9	
Age	Mean	43.2	48.8	54.3	60.3	
	Std. Dev.	13.7	13.8	12.2	13.4	
BMI	Mean	43.2	48.8	54.3	60.3	
	Std. Dev.	13.7	13.8	12.2	13.4	
Sodium intake	Mean	3.7	3.7	4.6	2.7	
	Std. Dev.	3.0	2.4	5.0	2.1	

Table 2 Polytomous logistic regression of a hypertension data: coefficient estimates and standard errors from GMWM and MLE.

Variable	Coefficients	MLE	GMWM	
		Estimates	Std. Err	p value	Estimates	Std. Err	p value	
Sex	β21	0.7062	0.2022	0.0002	1.3339	0.2269	<0.0001	
	β31	0.9789	0.3235	0.0012	1.0368	0.3013	0.0003	
	β41	1.4193	0.5746	0.0068	0.6753	0.2195	0.0010	
Age	β22	0.0350	0.0075	<0.0001	0.0671	0.0086	<0.0001	
	β32	0.0715	0.0121	<0.0001	0.1139	0.0133	<0.0001	
	β42	0.1096	0.0216	<0.0001	0.0753	0.0103	<0.0001	
BMI	β23	0.1147	0.0316	0.0001	0.1681	0.0360	<0.0001	
	β33	0.2422	0.0474	<0.0001	0.4382	0.0538	<0.0001	
	β43	0.4351	0.0884	<0.0001	0.2279	0.0388	<0.0001	
Sodium	β24	0.0104	0.0349	0.3829	0.1831	0.0355	<0.0001	
	β34	0.0919	0.0426	0.0155	0.2315	0.0486	<0.0001	
	β44	−0.2699	0.1580	0.9562	0.2294	0.0353	<0.0001	
Notes.

Std. Err, standard error.

The paper is set up as follows. In the next section we presents the basic notations, model, and standard GMM. “A robust GMM” introduces the outlier robust GMM estimator, and gives a detailed exposition of its implementation. In “Results”, we compares the performance of the standard MLE with the new estimator using a Monte-Carlo experiment. And we apply both estimators to real epidemiological data, and illustrate the usefulness of the robust estimator for application oriented researchers. We conclude with a discussion of advantages and limitations of the approach. The supporting document gathers the proofs of the asymptotic property.

Materials and Methods

The baseline-category logit model

Assume a random sample of size n from a large population. Each element in the population may be classified into one of J categories, denoted by yi=yi1,yi2,…,yiJ the multinomial trial for subject i, where yij=1 when the response is in category j and yij=0 otherwise, i=1,…,n, j=1,…,J. Thus, ∑jyij=1. Suppose p explanatory covariates, with at least one of them being continuous, are observed. Define xi=1,xi1,…,xip, and x=x1,…,xn. We assume that yi,xi are independently and identically distributed (i.i.d.). Let πij=πjxi=PYi=j|xi, denote the probability that the observation of Y belongs to category j, given covariates xi, we assume the relationship between the probability πj and x can be modeled as: (1) logπjxiπJxi=xiTβj,j=2,…,J

where βjT=βj0,βj1,…,βjp. Here we set the first category as reference class. This model is called a baseline-category logit model (Agresti, 2012) or generalized logit model (Stokes, Davis & Koch, 2009). MLE is usually used for obtaining parameter estimation of this model. Here we present an alternative estimation method formed with the GMM.

Estimation using GMM

The baseline-category logit model can be viewed as a multivariate model. Define yi∗T=yi2,…,yiJ, since yi1 is redundant. Let XT=X1T,…,XnT is a nJ−1×p+1J−1 matrix, with XiT, a J−1×p+1J−1 matrix, defined as: (2) XiT=xiTxiT⋯xiT.

In the GMM framework, we define (3) uβ=Xiyi∗−πi,i=1,…,n

where πiT=πi2,πi3,…,πiJ. And βT=β2T,β3T,…,βJT is the p+1J−1 vector of unknown parameters. The population moment condition is Euβ=0,

with the corresponding sample moment condition (4) Unβ=∑i=1nuβ.

The GMM estimation of βˆM can be obtained by minimizing the following quadratic objective function Qnβ=UnTβΣn−1βUnβ,

where Σnβ can be the empirical variance–covariance matrix given by Σnβ=1n2∑i=1nuTβuβ−1nUnβUnTβ.

Or, for the best efficiency of the GMM estimation, we can take the information matrix of the polytomous logit model (PLRM), that is, (5) Σnβ=∑i=1nXiDi−πiπiTXiT

where Di=diagonalπi.

In general, βˆM can be computed via an iterative procedure (Hansen, Heaton & Yaron, 1996). Under standard regularity conditions, the GMM estimator βˆM exists and converges in probability to the true parameter β0 (Hansen, 1982). A proof of asymptotic normality of GMM can be found on p. 2148 of Newey & McFadden (1994).

A robust GMM

In this section we introduce the outlier robust GMM estimator. In the following subsection, we specify moment conditions used for robust estimation. And the details on the implementation of the estimator follows.

The generalized method of weighted moments

The main principle used in the robust GMM estimator is that we replace moment conditions by a set of observation weighted moment conditions. Instead of Eq. (3), we define (6) uwβ=wiXiyi∗−πi−ci,i=1,…,n

where ci=EwiXiyi∗−πi. Then the estimation can be based on the moment conditions Euwβ=0.

Consequently, the generalized method of weighted moments (GMWM) estimates can be defined by (7) βˆw=argminβ∈BQnwβ

where (8) Qnwβ=UnwβTΣnwβ−1Unwβ,

with (9) Unwβ=∑i=1nuwβ.

Here we take the summation as the sample moment condition. The advantage of using the summation is that it can lead us to a direct estimation of covariance matrix.

It is clear to see that this definition is analogous to the standard GMM. If we choose wi=1 and ci=0 for all observations, the moment conditions in (6) are reduced to the standard moment conditions. Therefore, the standard GMM is a special case of the GMWM.

In order to specify the weights for the robust GMM estimator, we need the following definition of a distance, which is based on individual moment conditions: (10) diβ=uiwβTΣnwβ−1uiwβ,i=1,…,n.

The weight is assigned based on diβ, that is, wd=wdiβ. There are several alternative specifications of weight functions available in the literature (Huber, 1981; Hampel et al., 2005). In this study, the Huber’s weights are applied: (11) wdiβ=min1,cddiβ.

The above specification of weight function requires a value of the tuning constant cd. Both the outlier sensitivity and the efficiency of the estimator are determined by the constant. On the one hand, the estimator should be reasonably efficient if the sample contains no outlier. On the other hand, the estimator should be insensitive to outliers. To determine cd, understanding the distribution of diβ is critical. Clearly, uiwβ is a column vector, and diβ is a scalar quadratic distance, so we set cd=χ1−20.975/n, where χp−2⋅ is the quantile of the χ2 distribution with p degrees of freedom.

If we take the information matrix (5) of the PLRM as Σnwβ, we can compute leverage for each observation: (12) Hi=XiΣnwβ−1XiTσiw,i=1,…,n

where σiw is the ith component of Σnwβ. Then, a Mallows-type weight can be defined based on traceHi; that is, wx=wtraceHi, to downplay the observations with high leverages. Lesaffre & Albert (1989) suggest that the practical rule for isolating leverage points might set cx=2p+1J−1/n. In this study, we give observations with large leverages 0 weights, (13) wx=wtraceHi=1if traceHi≤2p+1J−1n0otherwise. 

An approach often used to combine the two weights is wi=wd⋅wx (Heritier et al., 2009).

The consistency correction vector ci is defined as ci=wdi1β−wdi0β/diagΣnwβ,i=1,…,n

where wdihβ=wXih−πiβ/diagΣnwβ−1 with h=0,1, is the weight for yi∗.

Implementation of the estimator

The continuous updating estimation method is applied in this study for estimating the regression coefficients and corresponding variance. The procedure is detailed as follows:

1. Apply an initial value β0 for computing Σnβ.

2. Compute diβ using Eq. (10) and Hi using Eq. (12); assign weights correspondingly based on (11) and (13).

3. With the combined weights, calculate Σnwβ and Unwβ in Eq. (9).

4. Obtain the estimator βˆw1 by minimizing Qnw of Eq. (8).

5. Go back to Step 1, replace β0 with the estimator βˆw1 in computing Σnwβˆw1, and move to the next iteration.

6. Continue this procedure until convergence criteria are met.

For the starting value β0, a reasonable choice is the MLE estimation based on the original data.

In the appendix, we proved that, under some regularity assumptions, we can have that βˆw is consistent for β0. And by studying the behavior of the weighted moment equations in a neighborhood of β0, we showed that the asymptotic linearity ensures the applicability of the central limit theorem for the asymptotic normality of GMWM.

Results

Monte Carlo simulations

In this section we investigate the properties of the GMWM estimator using a Monte-Carlo study. We generate data with three response categories and two covariates which are from multivariate normal distribution with 0 mean and identity covariance. The true coefficient matrix β0 is β0=β10β20β30β11β21β31β12β22β32=01.0−0.30−0.80.70−1.0−0.5.

Based on the specified coefficient values and using the probability based on the model (1), we compute the category-specific probabilities for each subject. Then, using the computed probabilities, we determine the most likely category to which each subject belongs. This decision is made through random generation from the multinomial distribution with the probability vector as a parameter. For instance, multinomial categories in R-Language are generated using rmultinormni,Ni,πxi function, where πxi=π1xi,…,πJxi is the probability vector, ni is the number of random vectors to draw, and Ni is the total number of objects that are put into J-categories. In our case, ni=Ni=1 for all subjects and J=3.

Two sample sizes, 100 and 1000, are examined. For each sample size, we run the simulation 1000 times. Average biases and MSEs are calculated and tabulated. Table 3 shows the results from randomly generated data with no outliers added. When the sample size is small, GMWM will give greater biases on β30 and β31 compared to the MLE method. For the sample size 1000, biases on these two parameters increase too, but not so obviously. Variances will also be inflated due to the weights we applied.

Table 3 Bias of parameter estimates and MSE from randomly generated data without outliers.

n	Parameter	True	MLE	GMWM	
Bias	MSE	Coverage	Bias	MSE	Coverage	
100	β20	1.0	0.0666	0.1030	0.945	0.0488	0.1986	0.949	
	β30	−0.3	−0.0059	0.1206	0.957	−0.1440	0.5578	0.952	
	β21	−0.8	−0.0654	0.1190	0.938	−0.0513	0.2550	0.961	
	β31	0.7	0.0566	0.1892	0.963	0.2318	0.5468	0.923	
	β22	−1.0	−0.0853	0.1764	0.969	−0.0691	0.2380	0.950	
	β32	−0.5	−0.0624	0.1453	0.945	0.0203	0.3195	0.964	
1000	β20	1.0	0.0050	0.0087	0.956	0.0043	0.0181	0.962	
	β30	−0.3	−0.0055	0.0105	0.984	−0.0106	0.0333	0.950	
	β21	−0.8	−0.0039	0.0099	0.943	−0.0013	0.0251	0.956	
	β31	0.7	0.0081	0.0160	0.968	0.0162	0.0401	0.954	
	β22	−1.0	−0.0071	0.0145	0.987	−0.0025	0.0258	0.948	
	β32	−0.5	−0.0047	0.0122	0.948	0.0041	0.0361	0.947	

Outliers are generated from a multivariate normal distribution with the mean vector =2,3 and identity covariance I2. For these outliers, their responses are intentionally misclassified, that is, they are placed within a different category from those predicted categories based on the true parameters.

Table 4 lists simulation results with outliers added. For estimations from datasets with 5% outliers, bias correction from the GMWM is excellent. However, when the datasets have 10% outliers, biases on estimations of some parameters (β21 and β22 in this simulation) are decreased, but not completely corrected.

Table 4 Comparison between GMWM and MLE estimation from randomly generated data with outliers added.

Size	Parameter	5% contamination	10% contamination	
		GMWM	MLE	GMWM	MLE	
		Bias	MSE	Coverage	Bias	MSE	Coverage	Bias	MSE	Coverage	Bias	MSE	Coverage	
100	β20	0.0568	0.1102	0.956	0.0860	0.0884	0.957	0.0489	0.0999	0.971	0.0868	0.0819	0.970	
	β30	−0.0038	0.1427	0.954	−0.0055	0.1528	0.949	−0.0057	0.1510	0.945	−0.0431	0.1461	0.814	
	β21	−0.0392	0.1464	0.949	0.2377	0.1360	0.785	0.0319	0.1227	0.946	0.3607	0.1933	0.579	
	β31	0.0175	0.2020	0.944	−0.1072	0.1270	0.921	−0.0235	0.1770	0.943	−0.1631	0.1283	0.949	
	β22	0.0374	0.1207	0.949	0.3848	0.2115	0.578	0.0207	0.0968	0.945	0.6088	0.4151	0.526	
	β32	−0.0548	0.1572	0.956	−0.0964	0.0904	0.964	−0.0817	0.1349	0.977	−0.1069	0.0803	0.967	
1000	β20	0.0172	0.0189	0.939	0.0490	0.0102	0.932	0.0451	0.0202	0.944	0.0657	0.0120	0.900	
	β30	0.0012	0.0340	0.945	0.0124	0.0075	0.952	−0.0071	0.0336	0.952	−0.0111	0.0063	0.822	
	β21	0.0260	0.0242	0.937	0.2874	0.0885	0.101	0.0164	0.0207	0.936	0.3876	0.1545	0.002	
	β31	−0.0058	0.0356	0.950	−0.1423	0.0345	0.697	−0.0497	0.0346	0.917	−0.2269	0.0658	0.521	
	β22	0.0366	0.0237	0.936	0.4390	0.2032	0.000	0.0238	0.0182	0.938	0.6500	0.4322	0.000	
	β32	−0.0106	0.0292	0.951	−0.0538	0.0103	0.940	−0.0434	0.0250	0.953	−0.0629	0.0106	0.902	

Application

For the hypertension data, the criterion for identifying observations with large distances is cd=0.22, and the criterion for identifying leverage points is cx=0.12. Applying the GMWM estimator, those blue-colored points in Fig. 1 are automatically downweighted, and red-colored points have 0 weight. The GMWM method indeed eliminates those inconsistencies: the coefficient of sodium intake for the odds model between the Stage 2 hypertension and the Normal categories is no longer negative, see the right side of Table 2.

As the results indicate, age, gender, and BMI all had significant impact on hypertension status. For example, one unit increase in BMI resulted in an increase of 1.26 (95% confidence interval [1.16–1.35]) times in likelihood to have Stage 2 hypertension when compared with the normal status. And with one year age increase, a subject was 1.07 (95% CI [1.06–1.10]) times more likely to have Stage 2 hypertension than to stay at the normal healthy status. Contrary to the MLE results for sodium intakes, which were difficult to make a conclusion due to inconsistent estimate, we now find that sodium intakes were statistically significant. When a daily intake of sodium increased one gram, a subject were 1.26 (95% CI [1.15–1.37]) times more likely to have Stage 1 hypertension, and 1.25 (95% CI [1.17–1.35]) times more likely to have Stage 2 hypertension. These results are consistent with the findings from previous studies (National Research Council, 2005; He & MacGregor, 2004).

Discussion

A reasonable choice to fit ordinal response data is the proportional odds model if the proportional odds assumption is not violated. Proportional odds models can take the ordinal information into modeling. And it reduces the number of parameters which is needed by the generalized logit model. Unfortunately, our data does not met the fundamental assumption of proportional odds models, which makes us choose to treat the outcome as a nominal response.

A datum with a nominal response and some continuous covariates is commonly seen in many scientific areas, such as sociology, economy, and biomedical studies. In order to be able to deal with outliers, we modified the GMM estimator to replace the standard moment conditions with weighted moment conditions, so that aberrant observations automatically receive less weight. We proved that the proposed method has good asymptotic behavior. When outliers are present, the GMWM estimator give much smaller biases than the estimations derived from the traditional MLE method. This method can be adapted to check whether results obtained with the traditional MLE approach are driven only by a few outlying observations. The weights produced from the robust procedure can be used to diagnose the cause of the differences and to indicate routes for model re-specification.

Appendix: Consistency and asymptotic normality

In this appendix, we introduce the assumptions for the asymptotic analysis of GMWM, and outline the derivations on the main asymptotic properties of GMWM.

We make the following sets of regularity assumptions regarding properties of the moment functions and identification assumptions. Assumption I I1. B is a compact parametric space.

I2. Σ is a positive definite matrix.

I3. It holds that Euwβ=0 if and only if β=β0, and for any ϵ>0, that infβ∈B∖Nβ0,ϵEuwβ>0

where Nβ0,ϵ=β∈Rl‖β−β0‖<ϵ is an open ϵ-neighborhood of a point β0.

Assumption F F1. Let uwβ be continuous in β∈B, and be twice differentiable in β on Nβ0,ϵ almost surely.

F2. Expectation E supβ∈Buwβ, E supβ∈Nβ0,ϵ∂uwβ/∂βk, and E supβ∈Nβ0,ϵ∂2uwβ/∂βk∂βl exists and are finite for k,l=1,…,p.

Assumption W W1. limϵ→0sup‖Δ‖≤ϵ|wβ+Δ−wβ|=0.

W2. limϵ→0sup‖Δ‖≤ϵ|∂wβ+Δ/∂β−∂wβ/∂β|=0.

When the above assumptions are met, we can prove that βˆw is consistent for β0. We begin with studying the behavior of the weighted moment equations in a neighborhood of β0. And proving their asymptotic linearity is followed. The linearity ensures the applicability of the central limit theorem for the asymptotic normality of GMWM. Theorem 1. Let the assumptions F and I hold, then the GMWM estimator βˆwis asymptotically normal, that is, nβˆw−β0⟶FN0,MTSwMas n→∞, whereM=VwTΣVw−1VwTΣ,

Sw=E1n∑i=1nuwβˆuwβˆT

with Vw=E∂Uwβˆ∂βT.

We start with proving two lemmas before we present the proof of Theorem 1.

Lemma 1. Let the assumptions F, I and W hold, and let Urwβ be the rth element of the vector Uwβ , r=1,…,p . Then, for 0<s<1 , (14) sup‖t‖≤C1n∑i∑l=1ptl∂/∂βlUi,rwβ+stn−∂/∂βlUi,rwβ=op1.

Proof. For l,r=1,…,p, by differentiating the ith component of Urwβ, we get ∂Ui,rwβ∂βl=−wβXi∂πiβ∂βl+∂wiβ∂βlXiyi−πiβ.

Then, sup‖t‖≤C1n∑i=1n∑l=1ptl∂/∂βlUi,rwβ+stn−∂/∂βlUi,rwβ≤Cn∑i=1n∑l=1ptlsup‖t‖≤C∂/∂βlUi,rwβ+stn−∂/∂βlUi,rwβ

and sup‖t‖≤C∂/∂βlUi,rwβ+stn−∂/∂βlUi,rwβ≤sup‖t‖≤Cwiβ+stn−wiβ∂/∂βlπiβ+stn+sup‖t‖≤C∂/∂βlπiβ+stn−∂/∂βlπiβ|Xiwiβ|+sup‖t‖≤C∂/∂βlwiβ+stn−∂/∂βlwiβXiyi−πiβ+stn+sup‖t‖≤Cyi−πiβ+stn−yi−πiβ∂/∂βlwiβ.

Then, by taking expectation at both sides, Esup‖t‖≤C∂/∂βlUi,rwβ+stn−∂/∂βlUi,rwβ≤sup‖t‖≤Cwiβ+stn−wiβsup‖t‖≤C∂/∂βlπiβ+stn+sup‖t‖≤C∂/∂βlπiβ+stn−∂/∂βlπiβsup‖t‖≤C|Xiwiβ|+sup‖t‖≤C∂/∂βlwiβ+stn−∂/∂βlwiβEsup‖t‖≤CXiyi−πiβ+stn+sup‖t‖≤Cyi−πiβ+stn−yi−πiβsup‖t‖≤C∂/∂βlwiβ.

Thus, by conditions F and W, we have Esup‖t‖≤C∂/∂βlUi,rwβ+stn−∂/∂βlUi,rwβ⟶0,∀i

and Esup‖t‖≤C1n∑i=1n∑l=1ptl∂/∂βlUi,rwβ+stn−∂/∂βlUi,rwβ⟶0,∀i.

Therefore, we have the results in (14).∎

Lemma 2. Let the assumptions F, I and W hold, it holds that (15) 1nsup‖t‖≤CUnwβ0+n−12t−Unwβ0+Vwn−12t=op1,

as n→∞ , where Vw=E∂Uwβ∂βT .

Proof. Write Unwβ0+n−12t−Unwβ0=∑i=1nwiβ0+n−12tuiβ0+n−12t−∑i=1nwiβ0uiβ0.

By the Taylor expansion, uiβ0+n−12t=uiβ0+n−12t∂∂βuiβ0+tn, where 0<s<1. Then, we can write (16) Unwβ0+n−12t−Unwβ0=∑i=1nuiβ0wiβ0+n−12t−wiβ0

(17) + 1n∑i=1nwiβ0t∂∂βuiβ0

(18) + 1n∑i=1nwiβ0+n−12t−wiβ0t∂∂βuiβ0

(19) + 1n∑i=1nwiβ0+tnt∂uiβ0+tn∂β−∂uiβ0∂β.

We will now show that terms (16), (18) and (19) are asymptotically negligible. As to the term (16), By assumption W1, wiβ0+n−12t−wiβ0→0, and uiβ0 is independent of β. So we have the term (16) tends to zero. Similarly, ∂/∂βuiβ0 is independent of β and t is bounded. Hence, the term (18) tends to zero. Lemma 1 implies ∂uiβ0+tn∂β−∂uiβ0∂β→0, as n→∞. So the term (19) can be neglect too.

Now, let us analyze the term (17). Let wi∗β0 be the limit of wiβ0. Rewrite (17) as (20) 1n∑i=1nwiβ0t∂∂βuiβ0=1n∑i=1nwiβ0−wi∗β0t∂∂βuiβ0

(21) +1n∑i=1nwi∗β0t∂∂βuiβ0−Ewi∗β0t∂∂βuiβ0

(22) +1nEwi∗β0t∂∂βuiβ0.

The first term (20) is negligible because ∂/∂βuiβ0 is independent of β, t is bounded, and wiβ0−wi∗β0→0. By the central limit theorem, each element of vector (21) converges in distribution to a normally distribution random variable with zero mean and a finite variance which is uniformly bounded by t. Hence, (21) is bounded in probability. The last term (20) is 1nEwi∗β0t∂∂βuiβ0=tnVw.

This proves the lemma.∎

Proof of Theorem 1. Since tn=n=Op1 as n→∞ by Lemma 2, we can write (15) as (23) Unwβ0+n−12tn−Unwβ0+Vwn−12tn=opn−12

with a probability arbitrarily close to one uniformly in tn∈t:‖t‖≤C. Moreover, with n−12tn=op1, ∂Unwβ0+n−12tn/∂β→Vw in probability as n→∞.

Note that the first order conditions of GMWM equal to 0, that is, ∂Qnwβw∂β=∂Unwβw∂βTΣβwUnwβw=0.

Replace Unwβw with Unwβ0+n−12tn from Eq. (23), ∂Unwβ0+n−12tn∂βTΣβwUnwβ0+n−12tn=Vw+op1TΣβwUnwβ0−Vwn−12tn=0.

Then we have (24) tn=nβw−β0=nVwTΣβwVw−1VwTΣβwUnwβ0+op1.

Next we examine the behavior of nUnwβ0, which can be written as (25) nUnwβ0=n−12∑i=1nuiβ0wiβ0=n−12∑i=1nuiβ0wiβ0−wi∗β0

(26) + n−12∑i=1nuiβ0wi∗β0.

Note that the term (25) is asymptotically negligible in probability due to the triangle inequality and assumption W1. The term (26) is a stationary sequence of absolutely random variables. By assumption I3 and F2, (26) have zero mean and finite second moments. So the central limit theorem can be applied on (26), giving nUnwβ0∼N0,Sw (Davidson, 1994, Section 25.3)

With Eq. (24), we have asymptotic normality of βw, and its asymptotic variance is given by MTSwM (Davidson, 1994).∎

Additional Information and Declarations

Competing Interests

Author Contributions

Xiaoshan Wang is an employee of the Center For Clinical And Translational Research, Forsyth Institute.

Xiaoshan Wang conceived and designed the experiments, performed the experiments, analyzed the data, contributed reagents/materials/analysis tools, wrote the paper, prepared figures and/or tables, reviewed drafts of the paper.

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
