# Peer review of "Modified generalized method of moments for a robust estimation of polytomous logistic model"

_PeerJ, doi:10.7717/peerj.467_

## Round 0.1 · original submission · Major Revisions

The reviewers raise important questions that need to be addressed; see below.

Reviewer 1 ·

Basic reporting

1. There are other weighted estimators, such as Kernel-weighted GMM estimator. You may want to refer to these estimators and compare with them in simulations.

2. In some applications, outlying continuous covariates are transformed into their log10 scales to remove outlying. How do you think of this alternative solution?

3. I think you did not finish section "Application", please check. It would be very interesting to present and compare coefficient estimates from your robust method and MLE in this section.

Experimental design

1. The motivating real dataset is a 2006 study on hypertension, in which the outcomes are ordinal: normal, pre-hypertension, stage I and stage II. Could you please give out more justifications for using polytomous logistic regression model instead of other models that specifically designed for ordinal outcomes, such as cumulative logits model? Sine polytomous logistic regression model is designed for both ordinal and nominal outcomes, the estimation efficiency from this model is not optimal.

2. I interpret covariate effects from polytomous logistic regression model as conditional, according to Equation (1). I think you need more explanations on the real data analysis part since you claim \textit{"... research objectives is to examine the association between hypertension and risk factors \textbf{in the population}"}. I recommend you giving more introductions and discussions on the polytomous logistic regression model.

3. I understand intuitively that the weight in Equation (11) will give observations with outliers smaller weights. Could you give intuitive explanations why these weights can also correct for outcome mis-classification?

4. In the part "the generalized method of weighted moments" between Equations (11) and (12), to justify the choice of tuning parameter c_d, you claimed Rank(u_i^w(\bm{\beta}))=1, I am not following this conclusion. You may want to define Rank and give some proof of this conclusion.

Validity of the findings

1. In the section "Discussion", you claim "the proposed method has good asymptotic behavior". I have seen the estimates are consistent from your simulations but how about their asymptotic distributions? You can include 95\% confidence interval coverage rates in Tables 3 and 4.

2. As for asymptotic behavior, could you please provide some theoretical references for the consistency and asymptotic distributions or give proof outlines?

3. In Table 2, I suggest you providing p.value instead of z-score.

·

Basic reporting

No Comments

Experimental design

No Comments

Validity of the findings

No Comments

Additional comments

This is a clearly written manuscript that describes a method to obtain robust estimation of polytomous logistic model. The approach is based on generalized method of moments. The author proved that the method works consistently. The only concern of mine is the starting values of parameters for the method. The author used the regular MLE estimates as starting values. When the data is contaminated, the regular MLE is biased. Such biased starting values might lead to convergence problem, especially when the starting values are seriously biased. It may need to find a better starting values.

---

## Round 0.2 · accepted · Accept

The paper can be published.

Reviewer 1 ·

Basic reporting

No Comments.

Experimental design

No Comments.

Validity of the findings

No Comments.

Additional comments

I would like to suggest the authors on the Appendix. They may want to check the modern empirical theorems for a "faster" proof.